# Development of Multiple Behaviors in Evolving Robots

Victor Massagué Respall [1] and Stefano Nolfi [1,2,*]

1   Institute of Robotics, Innopolis University, 420500 Innopolis, Russia; v.respall@innopolis.ru
2   Institute of Cognitive Sciences and Technologies, National Research Council, 00185 Roma, Italy
*   Correspondence: stefano.nolfi@cnr.it

**Abstract:** We investigate whether standard evolutionary robotics methods can be extended to support the evolution of multiple behaviors by forcing the retention of variations that are adaptive with respect to all required behaviors. This is realized by selecting the individuals located in the first Pareto fronts of the multidimensional fitness space in the case of a standard evolutionary algorithms and by computing and using multiple gradients of the expected fitness in the case of a modern evolutionary strategies that move the population in the direction of the gradient of the fitness. The results collected on two extended versions of state-of-the-art benchmarking problems indicate that the latter method permits to evolve robots capable of producing the required multiple behaviors in the majority of the replications and produces significantly better results than all the other methods considered.

**Keywords:** evolutionary robotics; multiple behaviors; multi-objective optimization

## 1. Introduction

Evolutionary robotics [1,2] is an established technique for synthesizing robots' behaviors that are difficult to derive analytically. The large majority of works carried in this area to date, however, focused on development of a single behavior only.

The capacity to exhibit multiple behaviors constitutes a key aspect of animal behavior and can play a similar important role for autonomous robots. Indeed, all organisms display a broad repertoire of behaviors. More precisely, in most of the cases the behavior of natural organism is organized in functionally specialized subunits governed by switch and decision points [3].

In this paper we investigate how standard evolutionary robotics methods can be extended to support the evolution of multiple behaviors.

The evolution of multiple behavior presents difficulties and opportunities. The difficulties originate from the fact that the processes that lead to the development of multiple behaviors can interfere among themselves. More specifically, the variations that are adaptive with respect to one behavior can be counter-adaptive with respect to another behavior. Consequentially, the retention of variations that are adaptive with respect to one behavior can reduce the ability to perform another required behavior. The opportunities originate from the fact that traits supporting the production of a given behavior can be reused to produce another required behavior [4] and consequently can facilitate the development of the latter behavior.

A possible way to reduce the problem caused by interferences consists in reducing the level of pleiotropy by fostering modular solutions. The term pleiotropy refers to traits that are responsible for multiple functions and or multiple behaviors. The hypothesis behind this approach is that the level of pleiotropy can be reduced by dividing the neural network controllers in modules responsible for the production of different behaviors since the variation affecting a module will tend to alter only the corresponding behavior [5–8]. Clearly, however, the reduction of pleiotropy also reduces the opportunities that can be gained from the possibility to re-use traits evolved for one behavior for the production of another required behavior [4]. In addition, neural modularity does not necessarily reduce

pleiotropy (see for example Calabretta et al. [5]). This can be explained by considering that the behavior of the robot is not simply a product of the brain of the robot. The behavior of a robot is a dynamical process that originate from the continuous interaction between the robot and the environment mediated by the characteristics of the brain and of the body of the robot. Consequently, there is not necessarily a one-to-one correspondence between neural modules and functional sub-units of the robot's behavior.

Several other studies investigate the role of modularity in classification and regression problems [9–11] and in the context of genetic regulatory networks [12,13]. However, the results obtained in these domains do not necessarily generalize to the evolution of embodied agents for the reasons described above.

A second possible strategy that can be used to reduce the problem caused by the interferences consists in using an incremental approach in which the robot is first encouraged to develop a first behavior and only later to develop additional behaviors, one at a time [14–16]. Eventually, the traits supporting the production of the first behaviors can be frozen during the development of successive behaviors, to avoid interferences. However, also this strategy reduces the opportunity for traits reuse. Indeed, the traits supporting the production of behaviors acquired during later stages cannot be reused for the production of behavior acquired earlier. Another weakness of this strategy is that it requires the intervention of the experimenter for the design and implementation of the incremental training process.

In this paper we explore a third strategy that consists in fostering the retention of variations that are adaptive with respect to all relevant behaviors. We evaluate this strategy in the context of standard evolutionary algorithms, that operate on the basis of selective reproduction and variation, and in the context of modern evolutionary strategies [17]. The latter algorithms operate by estimating the gradient of the expected fitness on the basis of the fitness collected and the variations received by individuals and by moving the population in the direction of the gradient. In the former case we foster the selection of variations adaptive to all behaviors by treating the performance on each behavior as separate objectives optimized through a multi-objective optimization algorithm [18,19]. In the latter case, we foster the selection of variations adaptive to all behaviors by calculating and using multi-objective fitness gradients.

The obtained results demonstrate how the usage of a modern evolutionary strategy combined with multi-objectives gradients permits to achieve very good results on state-of-the-art benchmark problems.

The efficacy of multi-objective optimization algorithms was already investigated in evolutionary robotics experiments involving fitness function with multiple components [20,21]. For example, in the case of robots evolved for the ability to navigate in an environment that are rewarded for: (i) the ability to move as fast as possible, (ii) the ability to move as straight as possible, and (iii) the ability to keep the activation of their infrared sensors as low as possible. Rather than computing the fitness on the basis of the sum of these three components, the components can be treated as separate objectives optimized through a multi-objective algorithm. Overall, these studies demonstrate that the usage of multi-objective optimization permits to evolve a more varied set of behaviors and reduce the probability to converge on local minima, in the case of experiments with multicomponent fitness functions. In the case of the first method proposed in this paper, instead, we applied a multi-objective optimization algorithm to the evolution of robots that should produce multiple behaviors. In our case, the objectives to be optimized correspond to the alternative fitness functions that are used in the contexts that require the exhibition of alternative behaviors.

The usage of evolutionary strategies that operate on the basis of multiple gradients was also investigated in previous studies [20,21]. This technique has been used to combine a true gradient, that is expensive to compute, with a surrogate gradient that constitute an approximation of the true gradient but that is easier to compute [21] or to combine the current gradient with historical estimated gradients [20]. In the case of the second method proposed in this paper, instead, we apply this method to evolution of multiple behaviors.

Consequently, we compute and use the gradients calculated with respect to the behaviors to be produced.

## 2. Method

To investigate the evolution of multiple behaviors we considered simulated neuro-robots evolved for the ability to produce two different behaviors in two different environmental conditions. We assume that the environmental conditions that indicate the opportunity to exhibit the first or the second behaviors are well differentiated. This is realized by including in the observation vector an "affordance" pattern that assume different values during episodes in which the robot should elicit the first or the second behavior. In the following sections we describe the adaptive problems, the neural network of the robots, and the evolutionary algorithms.

### 2.1. The Adaptive Problems

The problems chosen are an extended version of Pybullet locomotor problems [22]. These environments represent a free and more realistic implementation of the MuJoCo locomotor problems designed by Todorov, Erez and Tassa [23] and constitute a widely used benchmark for continuous control domains. We choose these problems since they are challenging and well-studied. The complexity of the problems is important, since the level of interference between the behaviors correlate with the complexity of the control rules that support the production of the required behaviors. Previous works involving situated agents that studied the evolution of multiple behaviors considered the following problems: (i) pill and ghost eating in a pac-man game [6], (ii) reaching a target position with a 2D three-segments arm [8,14] (iii) an inverted pendulum, a cart-pole balancing, and a single legged waling task [4], (iv) walking and navigation in simple multi-segments robots [15], (v) wheeled robot vacuum-cleaning an indoor environment [7], and (vi) wheeled robots provided with a 2 DOFs gripper able to find, pick-up and release cylinders [5].

The locomotors involve simulated robots composed by several cylindrical body parts connected through actuated hinge joints that can be trained for the ability to jump or walk toward a target destination as fast as possible. In particular, we selected the Hopper and the Ant problems. The Hopper robot has a single leg formed by a femur, a tibia and a foot that can jump (Figure 1, left). The Ant robot has a spherical torso and four evenly spaced legs formed by a femur and a tibia (Figure 1, right).

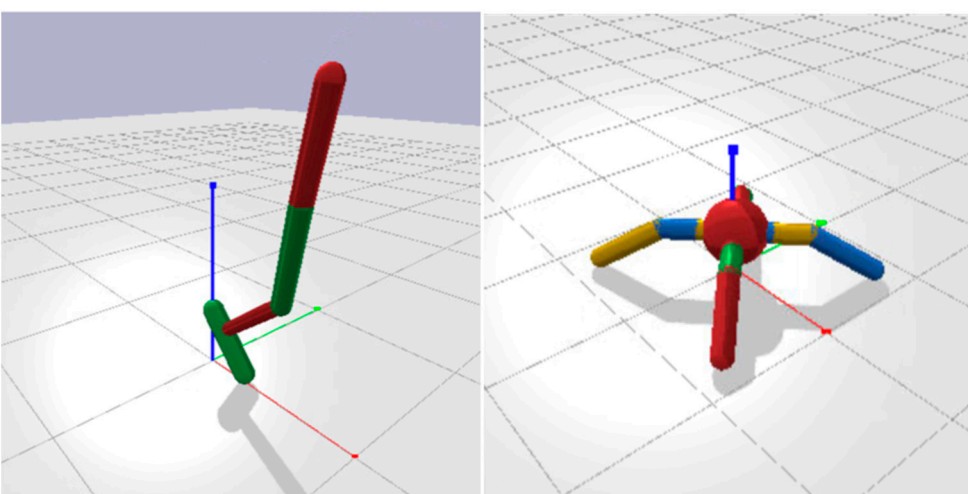

**Figure 1.** The Hopper (**left**) and the Ant (**right**).

In our extended version, the hopper is trained for jumping toward the target as fast as possible or for jumping vertically as high as possible while remaining in the same position.

The Ant is trained for the ability to walk 45 degrees left or right with respect to its frontal orientation.

In the case of the Hopper, this is realized by using the following fitness functions:

$$fitness_1 += \frac{d_t - d_{t-1}}{\Delta t} \tag{1}$$

$$fitness_2 += 2 \left| \frac{h_t - h_{t-1}}{\Delta t} \right| - 0.5 \left| \frac{d_t - d_{t-1}}{\Delta t} \right| \tag{2}$$

where *fitness*$_1$ and *fitness*$_2$ are the fitness functions used during the evaluation episode in which the agent should exhibit the first or the second behavior, respectively, *d* is the distance with respect to the target, *h* is the height of the torso with respect to the ground, and *t* is time.

In the case of Ant, we use the following fitness functions:

$$fitness_1 += \Delta p \, cos \left( \alpha - \frac{\pi}{4} \right) + 0.01 - (gl \times 0.1) - \left( a^2 \times 0.01 \right) \tag{3}$$

$$fitness_2 += \Delta p \, cos \left( \alpha + \frac{\pi}{4} \right) + 0.01 - (gl \times 0.1) - \left( a^2 \times 0.01 \right) \tag{4}$$

where fitness$_1$ and fitness$_2$ are the fitness functions used during the evaluation episode in which the agent should exhibit the first or the second behavior, $\Delta p$ is the Euclidean distance between the position of the torso on the plane at time *t* and *t$_{-1}$*, $\alpha$ is the angular offset between the frontal orientation of the Ant and the angle of movement during the current step, *gl* is the number of joint currently located on a limit, and *a* is the action vector (i.e., the activation of the motor neurons, see the next section). The bonus of 0.01 and the costs for the number of joints at their limits and for the square of the output torque are secondary fitness components that facilitates the evolution of walking behaviors (see Pagliuca, Milano and Nolfi [17]).

## 2.2. The Neural Network

The controller of the robot is constituted by a feedforward neural network with 17 and 30 sensory neurons (in the case of the Hopper and the Ant, respectively), 50 internal neurons, and 3 and 8 motor neurons (in the case of the Hopper and the Ant, respectively). The sensory neurons encode the orientation and the velocity of the robot, the relative orientation of the target destination, the position and the velocity of the joints, the contact sensors situated on the foot of the Hopper and on the terminal part of the four legs of the Ant, and the affordance vector. The affordance vector is set to [0.0 0.5] and to [0.5 0.0] during evaluation episodes in which the robots are rewarded with the first or the second fitness function illustrated above, respectively. The motor neurons encode the intensity and the direction of the torque applied by motors controlling the 3 and 8 actuated joints of the Hopper and of the Ant, respectively.

The internal and output neurons are updated by using tanh and linear activation functions, respectively. The state of the motor neurons is perturbed each step with the addition of Gaussian noise with mean 0.0 and standard deviation 0.01. The connection weights of the neural networks are encoded in free parameters and evolved. The number of connection weights is 1053 and 1958, in the case of the Hopper and of the Ant, respectively.

## 2.3. The Evolutionary Algorithms

We evolved the agents by using two state-of-the-art methods selected among standard evolutionary algorithms, that operate on the basis of selective reproduction and variation, and modern evolutionary algorithms, that estimated the gradient of the expected fitness on the basis of the fitness collected and the variations received by individuals and move the population in the direction of the gradient. Moreover, we designed and tested a variant

of each algorithm designed to enable the retention of variations producing progress with respect to all target behaviors.

The first method is the steady state algorithm (SSA) described in Pagliuca, Milano and Nolfi [24], see the pseudocode below (left). The procedure starts by creating a population of vectors that encode the parameters of a corresponding population of neural networks (line 1). Then, for a certain number of generations, the algorithm evaluates the fitness of the individuals forming the population (line 3), ranks the individual of the population on the basis of the average fitness obtained during two episodes evaluated with the two fitness functions (line 5), and replaces the parameters of the worse half individuals with varied copies of the best half individuals (lines 7–9). In 80% of the cases, the parameters of the new individuals are generated by crossing over each best individual with a second individuals selected randomly among the best half. The crossover is realized by cutting the vectors of parameters in two randomly selected points. In the remaining 20% of the cases, the parameters of the new individual are simply a copy of the parameter of the corresponding best individuals. (line 7). The parameters are then varied by adding a random Gaussian vector with mean 0.0 and variance 0.02 (line 8).

The variant Algorithm 1 designed for the evolution of multiple behaviors is the multi-objective steady state algorithm (MO-SSA), see the pseudocode below (right). In this case the ranking is made by ranking the individuals on the basis of the Pareto fronts to which they belong. The Pareto fronts are computed on the basis of the fitness obtained during the production of behavior 1 and 2 (line 5). The MO-SSS algorithm thus retain in the population the individuals that achieve the best performance with respect to behavior 1 or 2. This implies that the best individuals with respect to one behavior are retained even if they perform very poorly on the other behavior.

---

**Algorithm 1: designed for the evolution of multiple behaviors is the multi-objective steady state algorithm (MO-SSA).**

| SSA | MO-SSA |
|---|---|
| $\sigma = 0.02$: mutation variance | $\sigma = 0.02$: mutation variance |
| $\mu = 0.8$: crossover rate | $\mu = 0.8$: crossover rate |
| $\lambda = 40$: population size | $\lambda = 40$: population size |
| $\theta_{1-\lambda}$: population | $\theta_{1-\lambda}$: population |
| $f_n()$: fitness function for behavior n | $f_n()$: fitness function for behavior n |
| | |
| **1** initialize $\theta_\lambda$ : | **1** initialize $\theta_\lambda$ : |
| **2 for** $g = 1, 2, \ldots$ **do** | **2 for** $g = 1, 2, \ldots$ **do** |
| **3**   **for** $j = 1, 2, \ldots \lambda$ **do** | **3**   **for** $j = 1, 2, \ldots \lambda$ **do** |
| **4**     evaluate score: $s_i \leftarrow f_{12}(\theta_i)$ | **4**     evaluate score: $s_i \leftarrow f_{12}(\theta_i)$ |
| **5**   rank individuals by average fitness: u = ranks($s$) | **5**   rank individuals by pareto fronts: u = ranks($s$) |
| **6**   **for** $l = 1, 2, \ldots \frac{\lambda}{2}$ **do** | **6**   **for** $l = 1, 2, \ldots \frac{\lambda}{2}$ **do** |
| **7**     $\Phi = $ crossover( $\theta_{u[l]}$, $\theta_{u[rand]}$ ) or $\Phi = \theta_{u[l]}$ | **7**     $\Phi = $ crossover( $\theta_{u[l]}$, $\theta_{u[rand]}$ ) or $\Phi = \theta_{u[l]}$ |
| **8**     sample mutation vector: $\varepsilon \sim N(0, I) * \sigma$ | **8**     sample mutation vector: $\varepsilon \sim N(0, I) * \sigma$ |
| **9**     $\theta_{u[\frac{\lambda}{2}+l]} = \Phi + \varepsilon$ | **9**     $\theta_{u[\frac{\lambda}{2}+l]} = \Phi + \varepsilon$ |

---

The second method is the natural evolutionary strategy method (ES) proposed by Salimans et al. [25], see the pseudocode below (left). The algorithm evolves a distribution over policy parameters centered on a single parent $\theta$ composed of $\lambda2$ individuals. At each generation, the algorithm generates the gaussian vectors $\varepsilon$ that are used to perturb the parameters (line 4), and evaluate the offspring (lines 4, 5). To improve the accuracy of the fitness estimation the algorithm generates mirrored samples [26], i.e., generates $\lambda$ couples of offspring receiving opposite perturbations (lines 4, 5). The offspring are evaluated for two episodes for the ability to produce the two different behaviors (lines 5, 6). The average fitness values obtained during the two episodes are then ranked and normalized in the range $[-0.5, 0.5]$ (line 7). This normalization makes the algorithm invariant to the distribution of fitness values and reduce the effect of outliers. The estimated gradient g is then computed by summing the dot product of the samples $\varepsilon$ and of the normalized fitness

values (line 8). Finally, the gradient is used to update the parameters of the parent through the Adam [27] stochastic optimizer (line 9).

The variant Algorithm 2 designed for the evolution of multiple behaviors is the multi-objective evolutionary strategy (MO-ES), see the pseudocode below (right). In this case the algorithm compute two gradients (lines 3 and 9) by first evaluating the offspring for the ability to produce the behavior 1 and then behavior 2 (lines 6–7). The parameters of the parent are then updated by using the sum of the two gradients (line 10). The MO-ES algorithm thus moves the population in the directions that maximize the performance on both behavior 1 and 2, independently from the relative gain in performance that is obtained with respect to each behavior.

---

**Algorithm 2: The variant designed for the evolution of multiple behaviors is the multi-objective evolutionary strategy (MO-ES).**

| ES | MO-ES |
|---|---|
| $\sigma = 0.02$: mutation variance | $\sigma = 0.02$: mutation variance |
| $\lambda = 20$: half population size (total population size = 40) | $\lambda = 20$: half population size (total population size = 40) |
| $\theta$: policy parameters | $\theta$: policy parameters |
| $f_n()$: fitness function for behavior n | $f_n()$: fitness function for behavior n |
| optimizer = Adam | optimizer = Adam |
| | |
| **1** initialize $\theta_0$ | **1** initialize $\theta_0$ |
| **2 for** $g = 1, 2, \ldots$ **do** | **2 for** $g = 1, 2, \ldots$ **do** |
| | **3 for** b = 1, 2 |
| **3**   **for** $i = 1, 2, \ldots \lambda$ **do** | **4**   **for** $i = 1, 2, \ldots \lambda$ **do** |
| **4**     sample noise vector: $\varepsilon_i \sim N(0, I)$ | **5**     sample noise vector: $\varepsilon_i \sim N(0, I)$ |
| **5**     evaluate score: $s_i^+ \leftarrow f_{12}(\theta_{t-1} + \sigma * \varepsilon_i)$ | **6**     evaluate score: $s_i^+ \leftarrow f_b(\theta_{t-1} + \sigma * \varepsilon_i)$ |
| **6**     evaluate score: $s_i^- \leftarrow f_{12}(\theta_{t-1} - \sigma * \varepsilon_i)$ | **7**     evaluate score: $s_i^- \leftarrow f_b(\theta_{t-1} - \sigma * \varepsilon_i)$ |
| **7**     compute normalized ranks: u = ranks(s), $u_i \in [-0.5, 0.5]$ | **8**     compute normalized ranks: u = ranks(s), $u_i \in [-0.5, 0.5]$ |
| **8**   estimate gradient: $g_t \leftarrow \frac{1}{\lambda} \sum_{i=1}^{\lambda} (u_i * \varepsilon_i)$ | **9**   estimate gradient: $g_b \leftarrow \frac{1}{\lambda} \sum_{i=1}^{\lambda} (u_i * \varepsilon_i)$ |
| **9**   $\theta_g = \theta_{g-1} + \text{optimizer}(g)$ | **10**   $\theta_g = \theta_{g-1} + \text{optimizer}(g_1 + g_2)$ |

---

The evolutionary process is continued until total of $10^7$ evaluation steps are performed. The episodes last up to 500 steps and are terminated prematurely if the agents fall down. The initial posture of the agents is varied randomly at the beginning of each evaluation episode. The evolutionary process of each experimental condition is replicated 16 times.

The state of the actuators is perturbed with the addition of stochastic random noise with standard deviation 0.01 and average 0.0. The addition of noise makes the simulation more realistic and facilitates the transfer of solutions evolved in simulation in the real environment. The new methods proposed in this article do not alter the way in which the robots are evaluated with respect to standard method. Consequently, they do not alter the chance that the results obtained in simulation can be transferred in the real environment.

## 3. Results

Figures 2 and 3 display the average performance of the best agents evolved with the SSA and MO-SSA algorithms in the case of the Hopper and Ant problems, respectively (see the left side of the Figures). The video displaying the representative replications of the experiments are available online (see Section 5). As can be seen, the performance of the evolved robots is relatively good in the case of the Hopper (Figure 2) but rather poor in the case of the Ant (Figure 3). The performance obtained with the standard and multi-objective version of the algorithms, does not differ statistically, both in the case of the Hopper and in the case of the Ant (Mann–Whitney U test, *p*-value > 0.05).

To measure the fraction of agents capable to achieve sufficiently good performance during the exhibition of both behavior we post-evaluated the best evolved agents for 5 episodes on each behavior and we counted the fraction of agents that exceed a minimum

threshold on both behaviors. The evolved Hopper robots exceed a minimum threshold of 700 in 5/16 and 2/16 replications in the case of the SSA and MO-SSA algorithms, respectively (see Table 1). The evolved Ant robots exceed a minimum threshold of 400 in 0/16 and 0/16 replications in the case of the SSA and MO-SSA algorithms, respectively (see Table 2).

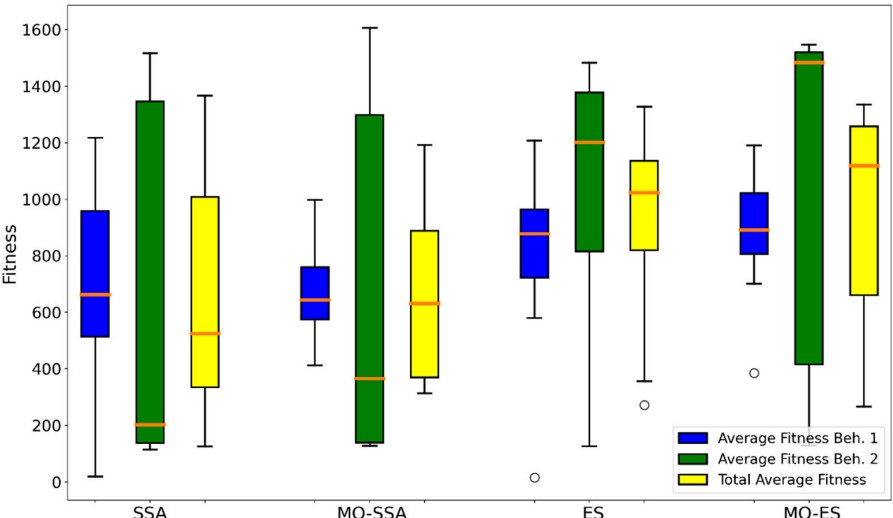

**Figure 2.** Performance of the best evolved agents obtained on the Hopper problem in the experiments performed with the SSA, MO-SSA, ES, and MO-ES algorithms. The blue, green, and yellow boxes show the fitness on the first behavior, on the second behavior, and on the two behaviors, respectively, obtained during a post-evaluation test in which the agents were evaluated for 5 episodes on each behavior. Data average over evaluation episodes. Boxes represent the inter-quartile range of the data and horizontal lines inside the boxes mark the median values. The whiskers extend to the most extreme data points within 1.5 times the inter-quartile range from the box. "o" indicates the outliers. See also Table 1.

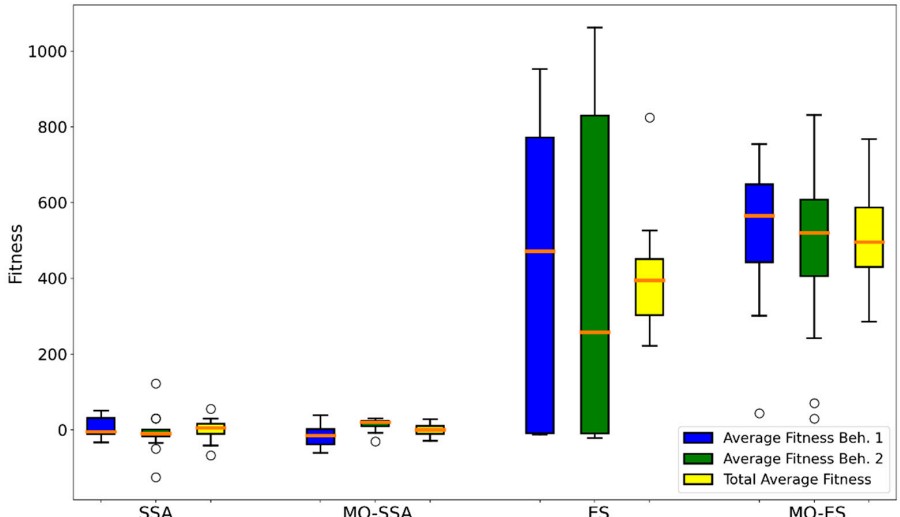

**Figure 3.** Performance of the best evolved agents obtained on the Ant problem in the experiments performed with the SSA, MO-SSA, ES and MO-ES algorithms. The blue, green, and yellow boxes show the fitness on the first behavior, on the second behavior, and on the two behaviors obtained during a post-evaluation test in which the robots were evaluated for 5 episodes on each behavior. Data average over evaluation episodes. The whiskers extend to the most extreme data points within 1.5 times the inter-quartile range from the box. "o" indicates the outliers. See also Table 2.

The variation of performance during the evolutionary process is shown in Figures 4 and 5 (top Figures).

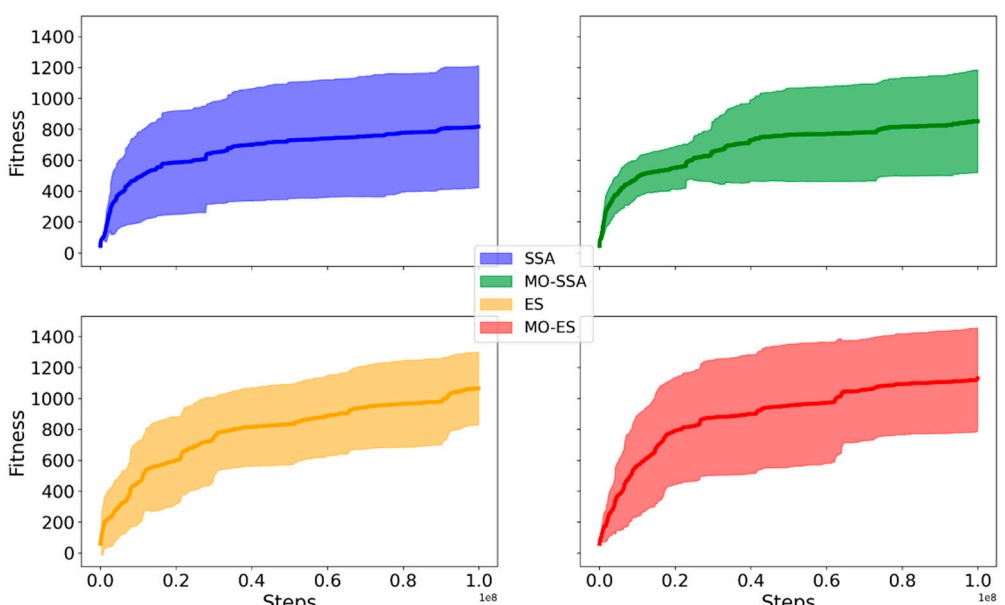

**Figure 4.** Performance of the best evolved agents during the evolutionary process in the case of the Hopper problem. Results obtained with the SSA (top-left), MO-SSA (top-right), ES (bottom-left) and MO-ES (bottom right) algorithms. Average results of the best agents of successive generations post-evaluated for four episodes for the ability to produce the two behaviors. Shadows indicate the standard deviation.

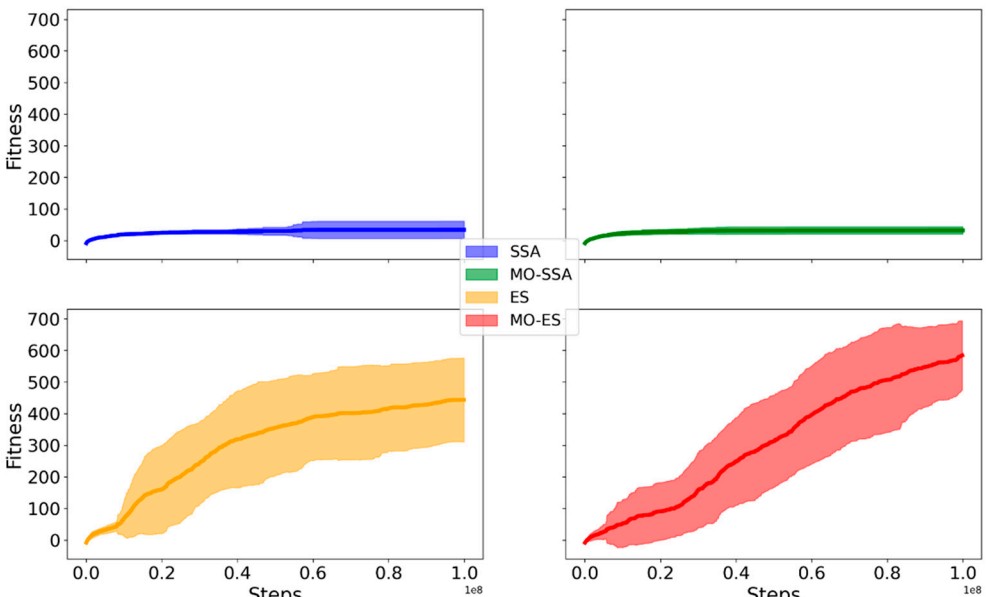

**Figure 5.** Performance of the best evolved agents during the evolutionary process in the case of the Ant problem. Results obtained with the SSA (top-left), MO-SSA (top-right), ES (bottom-left) and MO-ES (bottom right) algorithms. Average results of the best agents of successive generations post-evaluated for 4 episodes for the ability to produce the two behaviors. Shadows indicate the standard deviation.

The videos displaying a representative replication of the experiments are available online (see Section 5). As can be seen, the performance of the evolved robots is quite good

both in the case of the Hopper (Figure 2) and the Ant (Figure 3), and is significantly better than the performance obtained with the SSA and MO-SSA algorithms (Mann-Whitney U test with Bonferrori correction, *p*-value < 0.05). The MO-ES algorithm is significantly better than the ES method (Mann-Whitney U test, *p*-value < 0.05).

The evolved Hopper robots exceed a minimum threshold of 700 in 10/16 and 11/16 replications in the case of the ES and MO-ES algorithms, respectively (see Table 1). The evolved Ant robots exceed a minimum threshold of 400 in 1/16 and 10/16 replications in the case of the ES and MO-ES algorithms, respectively (see Table 2).

The variation of performance during the evolutionary process is shown in Figures 4 and 5 (bottom side). As can be seen, in the case of the Hopper the MO-ES algorithm outperform the ES algorithm from the beginning of the evolutionary process. In the case of the Ant, instead, the MO-ES algorithm outperform the ES algorithm during in the second half of the evolutionary process.

**Table 1.** Performance of the best evolved agents obtained on the Hopper problem in the experiments performed with the SSA, MO-SSA, ES and MO-ES algorithms obtained in each replication of the experiment. The columns show the data obtained in different replications. The top, central and bottom tables show the average performance during 5 episodes in which the robot should perform the first behavior, the average performance during 5 episodes in which the robot should perform the second behavior, and the average performance during 10 episodes in which the robot should perform the two behaviors. The numbers in bold indicate the replications in which the agents exceed the threshold of 700 on both behaviors.

| | Behavior 1 | | | |
|---|---|---|---|---|
| | **SSA** | **MO-SSA** | **ES** | **MO-ES** |
| **S1** | **1096.72** | 412.44 | 1069.40 | 892.48 |
| **S2** | 737.69 | 501.08 | 580.50 | **701.61** |
| **S3** | 545.12 | 646.52 | **1049.04** | **824.20** |
| **S4** | 421.28 | **797.60** | 622.18 | **871.90** |
| **S5** | 678.52 | 997.87 | **935.51** | 754.43 |
| **S6** | 577.79 | 640.83 | **851.71** | **889.58** |
| **S7** | 499.99 | 864.03 | **862.67** | **1012.49** |
| **S8** | **1217.26** | 797.76 | 15.90 | 974.87 |
| **S9** | 482.29 | 691.44 | **1147.98** | **1050.4** |
| **S10** | 1076.87 | 499.17 | **1207.67** | **722.74** |
| **S11** | **709.25** | 704.96 | 893.43 | **1191.21** |
| **S12** | **944.09** | 612.05 | **797.51** | 891.43 |
| **S13** | 19.34 | **747.36** | **756.36** | **1150.51** |
| **S14** | 519.64 | 575.41 | **915.07** | **965.83** |
| **S15** | **1004.87** | 570.44 | **905.41** | 385.46 |
| **S16** | 648.29 | 599.08 | 604.81 | **1093.81** |
| **Avg.** | 698.69 | 666.13 | 825.95 | 898.31 |
| **Std.** | 298.32 | 145.32 | 274.29 | 192.64 |

**Table 1.** *Cont.*

| | Behavior 2 | | | |
| --- | --- | --- | --- | --- |
| | SSA | MO-SSA | ES | MO-ES |
| **S1** | **1473.99** | 1230.24 | 601.06 | 144.96 |
| **S2** | 173.82 | 150.34 | 131.38 | **1522.57** |
| **S3** | 133.88 | 994.05 | **1469.5** | **1402.38** |
| **S4** | 139.68 | **1585.82** | 926.03 | **1521** |
| **S5** | 139.07 | 200.96 | **1260.08** | 131.02 |
| **S6** | 803.13 | 1539.3 | **1360.06** | **1507.63** |
| **S7** | 1508.83 | 138.44 | **1351.32** | **1496.32** |
| **S8** | **1516.36** | 532.39 | 528.31 | 129.23 |
| **S9** | 130.09 | 1502.92 | **1349.72** | **1489.18** |
| **S10** | 114.10 | 128.39 | **1446.06** | 1525.08 |
| **S11** | **1318.39** | 130.25 | 126.32 | **1480.15** |
| **S12** | **1067.41** | 128.61 | **1431.69** | 504.44 |
| **S13** | 232.47 | **1605.82** | 1143.55 | **1519.1** |
| **S14** | 124.54 | 948.71 | **1483.15** | **1246.8** |
| **S15** | **1430.51** | 143.40 | **886.25** | 149.01 |
| **S16** | 142.74 | 139.95 | 1118.33 | **1547.13** |
| **Avg.** | 653.06 | 693.72 | 1037.68 | 1082.25 |
| **Std.** | 596.89 | 606.17 | 447.40 | 596.43 |
| | Behavior 1 and 2 | | | |
| | SSA | MO-SSA | ES | MO-ES |
| **S1** | **1285.36** | 821.34 | 835.23 | 518.72 |
| **S2** | 455.75 | 325.71 | 355.94 | **1112.09** |
| **S3** | 339.50 | 820.28 | **1254.27** | **1113.29** |
| **S4** | 280.48 | **1191.71** | 774.27 | **1196.45** |
| **S5** | 408.80 | 599.42 | **1097.80** | 442.73 |
| **S6** | 690.46 | 1090.07 | **1105.88** | **1198.61** |
| **S7** | 1004.41 | 501.24 | 1107.00 | **1254.40** |
| **S8** | **1366.81** | 665.08 | 272.11 | 552.05 |
| **S9** | 306.19 | 1097.18 | **1248.85** | **1269.79** |
| **S10** | 595.49 | 313.78 | **1326.86** | **1123.91** |
| **S11** | **1013.82** | 417.61 | 509.88 | **1335.68** |
| **S12** | **1005.75** | 370.33 | **1114.60** | 697.94 |
| **S13** | 125.91 | **1176.59** | **949.96** | **1334.81** |
| **S14** | 322.09 | 762.06 | **1199.11** | **1106.31** |
| **S15** | **1217.69** | 356.92 | **985.83** | 267.24 |
| **S16** | 395.52 | 369.51 | 861.57 | **1320.47** |
| **Avg.** | 675.88 | 679.93 | 931.81 | 990.28 |
| **Std.** | 396.25 | 312.34 | 310.2 | 350.41 |

**Table 2.** Performance of the best evolved agents obtained on the Ant problem in the experiments performed with the SSA, MO-SSA, ES and MO-ES algorithms obtained in each replication of the experiment. The columns show the data obtained in different replications. The top, central and bottom tables show the average performance during 5 episodes in which the robot should perform the first behavior, the average performance during 5 episodes in which the robot should perform the second behavior, and the average performance during 10 episodes in which the robot should perform the two behaviors. The numbers in bold indicate the replications in which the agents exceed the threshold of 400 on both behaviors.

| | **Behavior 1** | | | |
|---|---|---|---|---|
| | **SSA** | **MO-SSA** | **ES** | **MO-ES** |
| **S1** | 28.13 | −60.50 | −8.040 | 540.68 |
| **S2** | −9.12 | 25.20 | −11.25 | **623.29** |
| **S3** | −11.05 | −6.78 | 878.46 | **420.64** |
| **S4** | 37.99 | 29.82 | −2.91 | 43.92 |
| **S5** | −9.52 | −23.28 | 477.99 | **562.64** |
| **S6** | 37.64 | −26.89 | **952.42** | **570.57** |
| **S7** | 50.17 | −42.53 | −12.58 | **435.61** |
| **S8** | −33.00 | 38.87 | 767.71 | **703.39** |
| **S9** | −9.54 | −47.19 | −9.11 | 713.02 |
| **S10** | −11.14 | −29.84 | −9.95 | **630.31** |
| **S11** | 50.29 | −2.6 | 650.67 | 301.46 |
| **S12** | −11.09 | 14.83 | 927.54 | 567.45 |
| **S13** | 29.16 | −1.86 | 504.61 | **735.10** |
| **S14** | −14.75 | −49.52 | 785.75 | **444.53** |
| **S15** | 19.02 | −6.02 | 464.74 | **754.73** |
| **S16** | −0.91 | −36.76 | −9.36 | 556.15 |
| **Avg.** | 8.86 | −14.07 | 396.67 | 537.72 |
| **Std.** | 25.75 | 29.41 | 383.36 | 175.99 |
| | **Behavior 2** | | | |
| | **SSA** | **MO-SSA** | **ES** | **MO-ES** |
| **S1** | 30.46 | 26.63 | 893.51 | 29.78 |
| **S2** | −125.04 | −4.91 | 945.64 | **422.23** |
| **S3** | −9.20 | 10.60 | −10.09 | **530.32** |
| **S4** | −8.73 | 18.75 | 640.8 | 658.6 |
| **S5** | −11.68 | 21.33 | −9.82 | **432.32** |
| **S6** | −10.25 | −30.98 | **696.7** | **598.37** |
| **S7** | −9.86 | 23.69 | 522.48 | **637.71** |
| **S8** | −49.88 | 16.13 | −10.46 | **831.25** |
| **S9** | 30.71 | 21.93 | 1061.68 | 70.37 |
| **S10** | 122.43 | 28.49 | 808.8 | **680.71** |
| **S11** | −10.67 | 24.41 | −9.04 | 576.28 |
| **S12** | −34.99 | 21.61 | −9.11 | 241.57 |
| **S13** | −33.99 | 6.95 | −10.05 | **452.29** |

**Table 2.** *Cont.*

| | | | | |
|---|---|---|---|---|
| **S14** | 29,47 | 30.28 | −6.35 | **545.87** |
| **S15** | −10.82 | 12.31 | −21.73 | **510.67** |
| **S16** | −8.31 | −8.10 | 905.97 | 357.41 |
| **Avg.** | −6.90 | 13.70 | 399.31 | 473.48 |
| **Std.** | 48.91 | 15.75 | 426.73 | 209.10 |
| | **Behavior 1 and 2** | | | |
| | **SSA** | **MO-SSA** | **ES** | **MO-ES** |
| **S1** | 29.29 | −16.94 | 442.74 | 285.23 |
| **S2** | −67.08 | 10.14 | 467.20 | **522.76** |
| **S3** | −10.12 | 1.91 | 434.19 | **475.48** |
| **S4** | 14.63 | 24.28 | 318.95 | 351.26 |
| **S5** | −10.60 | −0.98 | 234.08 | **497.48** |
| **S6** | 13.70 | −28.94 | **824.56** | **584.47** |
| **S7** | 20.16 | −9.42 | 254.95 | **536.66** |
| **S8** | −41.44 | 27.5 | 378.62 | **767.32** |
| **S9** | 10.59 | −12.63 | 526.28 | 391.69 |
| **S10** | 55.65 | −0.68 | 399.43 | **655.51** |
| **S11** | 19.81 | 10.91 | 320.82 | 438.87 |
| **S12** | −23.34 | 18.22 | 459.22 | 404.51 |
| **S13** | −2.42 | 2.52 | 247.28 | **593.69** |
| **S14** | 7.36 | −9.62 | 389.7 | **495.2** |
| **S15** | 4.10 | 3.15 | 221.31 | **632.7** |
| **S16** | −4.61 | −22.43 | 488.31 | 456.78 |
| **Avg.** | 0.98 | −0.19 | 397.99 | 505.6 |
| **Std.** | 27.76 | 15.59 | 142.88 | 118.81 |

## 4. Discussion

We investigated how standard evolutionary robotics methods can be extended to support the evolution of multiple behaviors. More specifically we investigated whether forcing the retention of variations that are adaptive with respect to all required behaviors facilitate the concurrent development of multiple behavioral skills.

We considered both standard evolutionary algorithms, in which the population is formed by varied copies of selected individuals, and modern evolutionary strategies, in which the population is distributed around a single parent and in which the parameters of the parent are moved in the direction of the gradient of the expected fitness. The retention of variations adaptive with respect to all behaviors should be realized in a different way depending on the algorithm used. In the case of standard evolutionary algorithms, it can be realized by using a multi-objective optimization technique, i.e., by selecting the individuals located in the first Pareto fronts of the multidimensional space of the fitness of multiple behaviors. In the case of modern evolutionary strategies, it can be realized by computing multiple gradients and by moving the center of the population in the directions corresponding to the vector sum of the gradients. This method to pursue multi-objective optimization in evolutionary strategies is original, as far as we know.

We evaluated the efficacy of the two methods on two extended versions of the Hopper and Ant Pybullet locomotor problems in which the Hopper is evolved for the ability to jump toward a target destination as fast as possible or to jump on the place as high as

possible and in which the Ant is evolved for the ability to walk 45 degrees left or right with respect to its current orientation.

The obtained results indicate that Salimans et al. [25] evolutionary strategy extended with multiple gradients calculation permits to obtain close to optimal performance for both problems. The performance obtained are statistically better than the control condition that rely on a single gradient and statistically better than the results obtained with the other algorithm considered. Moreover, the analysis of the evolved robots demonstrate that they manage to display sufficiently good performance on both behaviors in most of the replications.

In the case of the standard algorithm, instead, the selection of the individuals located on the first Pareto fronts does not produce better performance with respect to the control condition in which the robots are selected on the basis of the average performance obtained during the production of the two behaviors. The analysis of the evolved robots indicate that they are able to achieve sufficiently good performance on both behaviors only in a minority of the replications in the base of the Hopper and in none of the replications in the case of the Ant.

## 5. Conclusions

We introduced a variation of a state-of-the-art evolutionary strategy [25] that support the evolution of multiple behaviors in evolving robots. The new MO-ES algorithm moves the population by using the vector sum of the gradients of the expected fitness computed with respect to each behavior. The obtained results demonstrated that the method is effective and produce significantly better results than the standard ES algorithm. This new method also outperforms significantly the other two algorithms considered: a standard steady state algorithm (SSA) and a multi-objective steady state algorithm (MO-SSA) that operates by selecting the individuals located on the first pareto-fronts of the objectives of the behaviors.

The relative efficacy of the algorithm proposed with respect to alternative methods remain to be investigated in future works. Carrying out a quantitative comparison can result difficult, due to the specificity of the requirements imposed by each method, but can provide valuable insights.

A second aspects that deserves future investigation is the scalability of the method proposed with respect to the number of behaviors and to the complexity of the behaviors.

## 6. Online Resources

The videos displaying the behaviors of the best robots evolved in each condition are available from the following links:

| Problem | Algorithm | Video |
|---------|-----------|-------|
|         | SSA       | https://youtu.be/uOFX266aurA |
| Hopper  | MO-SSA    | https://youtu.be/fsyybqX7XUM |
|         | ES        | https://youtu.be/B-xgwY2aC6g |
|         | MO-ES     | https://youtu.be/6Z8jElYOYYY |
|         | SSA       | https://youtu.be/thaD5x2aKH8 |
| Ant     | MO-SSA    | https://youtu.be/g8Un3LLXgVc |
|         | ES        | https://youtu.be/-j2Sy6jHSNM |
|         | MO-ES     | https://youtu.be/ptxbcPDdxiE |

**Author Contributions:** Conceptualization, S.N.; Data curation, V.M.R.; Formal analysis, V.M.R.; Investigation, S.N.; Methodology, S.N.; Software, V.M.R. and S.N.; Supervision, S.N.; Validation, V.M.R.; Writing—original draft, V.M.R. All authors have read and agreed to the published version of the manuscript.

**Funding:** This research received no external funding.

**Data Availability Statement:** Data is contained within the article or supplementary material.

**Acknowledgments:** Stefano Nolfi thanks Ángel García-Casillas del Riego for insightful discussion.

**Conflicts of Interest:** The authors declare no conflict of interest.

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
