# Peer review of "Development of Multiple Behaviors in Evolving Robots"

_robotics, doi:10.3390/robotics10010001_

Round 1

Reviewer 1 Report

The manuscript presents an approach to the problem of interferences in the processes of using evolutionary robotics methods to generate multiple behaviours. The approach consists in fostering the retention of variations that are adaptive with respect to all relevant behaviours. Two evolutionary robotics algorithms were used with a hopper agent and with an ant agent in simulation. One of the algorithms is steady state algorithm (SSA) and the other algorithm is natural evolutionary strategy method (ES). For each algorithm, a variant was designed to enable the retention of variations to generate multiple behaviours (MO-SSA and MO-ES respectively). Performance of SSA, MO-SSA, ES, MO-ES was compared in the two simulated agents. The concluded that the retention of the variations that are adaptive with respect to all behaviors enables the development of multiple behaviours.

The manuscript is relatively easy to follow given it is structured content. However, the work lacks major points given its current state.

First, from Figure 2-5, it appears the benefit of having retention is obvious in Figure 5, but not in other cases. For example, median value, box and whisker ranges are similar in Figure 2; neither is satisfactory in Figure 3; and median value is higher but box and whisker ranges are larger in Figure 4. How did the author come to the conclusion that the proposed approach gives added value?

Second, given the three approaches to reduce interferences, how does the proposed approach compare to the other two approaches? Some quantitative comparison would be needed.

Third, one of the challenges of evolutionary robotics methods is to bring the gap between the simulation and the real-world robots. Discussion is needed to a provide assumptions, plausibility or hurdles when translate the method to real world.

Fourth, the numbers reported in the text seem to be misaligned with the values indicated in the figures. For example, on page 6 line 193: “The evolved Hopper robots exceed a minimum threshold of 700 in 5/16 and 2/16 replications in the case of the SSA and MO-SSA algorithms, respectively.” Figure 2 shows median value just above 700 and lower box range of just above 400 and just above 550. Please confirm all numbers in the text match those in Figure 2-5.

Minor comments,

Fifth, page 7 line 222, “SSA and MO-SSA” seems to refer to “ES and MO-ES”?

Sixth, the paragraph of “Average performance obtained during a post-evaluation test in which the robots were evaluated for 4 episodes for the ability to produce the two behaviors. … Data obtained by running 16 replications of each experiment.” Are repeated three times. Keep one and move it the beginning of the Results section.

Author Response

First, from Figure 2-5, it appears the benefit of having retention is obvious in Figure 5, but not in other cases. For example, median value, box and whisker ranges are similar in Figure 2; neither is satisfactory in Figure 3; and median value is higher but box and whisker ranges are larger in Figure 4. How did the author come to the conclusion that the proposed approach gives added value?

In addition to plotting the distribution of the results we analyzed weather the result differ statistically. As stated in the manuscript, the performance of the SSA and MO-SSA displayed in Figure 2 and 3 do not differ statistically both in the case of the Hopper and of the Ant (Mann-Whitney U test, p-value > 0.05). The performance of the ES and MO-ES algorithm differ statistically both in the case of the Hopper and of the Ant (Mann-Whitney U test, p-value < 0.05), as reported in the manuscript. This demonstrates that the MO-ES algorithm, proposed in the paper, provides an advantage. The performance of the MO-ES algorithm also differs statistically from the performance of the SSA and the MO-SSA algorithms. An additional measures that show the advantage of the MO-ES method over alternative methods is that the robots evolved with the MO-ES display sufficiently good performance during the exhibition of both behaviors in the majority of the replications both in the case of the Hopper and the Ant. The other methods, instead, achieve sufficiently good performance in only few replications or manage to achieve sufficiently good performance in the case of the Hopper (the simpler problem) but not also in the case of the Ant (the most complex problem).

Second, given the three approaches to reduce interferences, how does the proposed approach compare to the other two approaches? Some quantitative comparison would be needed.

We expanded the description of the related literature and of the advantages of our method with respect to alternative strategies (i.e. modular, and incremental).

The objective of the manuscript is that to introduce and analyze experimentally a new method that can promote the development of multiple behaviors. Benchmarking qualitatively different methods is certainly an interesting issue but it is outside the scope of this manuscript. We included a concluding section in which we mention the quantitative comparison with other methods within future works. Please also consider that a quantitative comparison of this type would require months of works and would require to figure out ways to overcome the problems caused by the presence of difference constraints.  For example, incremental approaches, require the usage of domain knowledge to shape the course of the adaptive process, unlike our method. The utilization of this knowledge in one method but not in the other will make the comparison unfair.

Third, one of the challenges of evolutionary robotics methods is to bring the gap between the simulation and the real-world robots. Discussion is needed to a provide assumptions, plausibility or hurdles when translate the method to real world.

The new methods proposed do not alter the way in which the robots are evaluated with respect to standard method. Consequently, they do not alter the chance that the results obtained in simulation can be transferred in the real environment. We specified that at the end of Section 2 of the revised manuscript. We also clarified that we add noise to the actuator to make the simulation more realistic and to facilitate the transfer of robots evolved in simulation in hardware.

Fourth, the numbers reported in the text seem to be misaligned with the values indicated in the figures. For example, on page 6 line 193: “The evolved Hopper robots exceed a minimum threshold of 700 in 5/16 and 2/16 replications in the case of the SSA and MO-SSA algorithms, respectively.” Figure 2 shows median value just above 700 and lower box range of just above 400 and just above 550. Please confirm all numbers in the text match those in Figure 2-5.

This is due to the fact that the data displayed in Figure 2-5 represent the average performance obtained during the exhibition of the two behaviors. The data reported in the text that describe the number of replications in which the robots exceed a minimum threshold level on both behaviors, instead, refer to the number of evolved robots that exceed the threshold during the production of both behaviors. Consider for example two Ant experiments that produce a performance of [450, 550] and [350,650] on the two behaviors. The average performance is 500 in both cases. However, only the first experiment produces sufficiently good performance on both behaviors (i.e. a performance equal or greater than 400 both during the production of the first and during the production of the second behavior)

To make this clearer, we specified explicitly in the figure captions of the revised manuscript that the data indicate the average performance on the two behaviors.

Minor comments,

Fifth, page 7 line 222, “SSA and MO-SSA” seems to refer to “ES and MO-ES”?

Fixed. (line 242)

Sixth, the paragraph of “Average performance obtained during a post-evaluation test in which the robots were evaluated for 4 episodes for the ability to produce the two behaviors. … Data obtained by running 16 replications of each experiment.” Are repeated three times. Keep one and move it the beginning of the Results section.

Fixed

Reviewer 2 Report

The paper investigate evolutionary robotics techniques can be improved to support the evolution of multiple behaviours by including the retention of variations that are adaptive with respect to all required behaviours. The authors considered traditional evolutionary algorithms, where the population is formed by adapted copies of selected individuals, and an evolutionary strategies, where the population is distributed around an individual (parent) and the parameters of the individual are varied in the direction of the gradient of the expected fitness. 

The investigation are performed using two simulated robot structures, a hopper robot and an ant robot. Both having different performance (fitness) measures. A relatively small number of experiments are performed for both these robots to show the efficacy of the ideas outlined in the paper.

The results presented in the paper indicate that the evolutionary strategy highlighted and extended with multiple gradients produces good performance for both problems. The authors suggest (and it appear to be correct) that the performance obtained are better than solutions that rely only on a single gradient method. 

The paper does demonstrates that the use of variations that are adaptive with respect to all behaviours do enable the development of multiple behaviours.

While the writing quality of the paper is okay, I do feel it could be improved enough for extra effort to be made for such improvements. 

Author Response

While the writing quality of the paper is okay, I do feel it could be improved enough for extra effort to be made for such improvements.

We included additional reference to related works. We expanded the description of the related literature and of the relation between our method and other related research. We expanded the description of the motivation behind the chosen experimental scenario. We expanded the description of the implications of our work. We included a description of relevant future works.

Reviewer 3 Report

The paper presents a comparative study regarding different strategies for learning multiple behaviours based on evolutionary robotics and multiobjective optimization.  

The text is well organised, contributions are relevant and methods are adequate. However, the proposal was made based on a limited literature review. This must be extended to describe not only the algorithms employed on related work but also the experimental scenarios used and how did they motivate the experiments proposed in this work. Results and analyses from other studies should also be discussed.

I would also suggest a more detailed analysis of the evolution process, including the average fitness over time. Besides, the box-plots used to present the results could be summarised on a single figure, allowing a more straight-forward comparison between the different architectures considered. Some outliers should also be removed for better clarity.  

Overall, I expect a more in-depth analysis of state-of-the-art, methodology and results for the paper to be accepted as a contribution to this journal. Not only on the motivation for the work carried out including a more comprehensive literature review, but also on the analysis and presentation of results. A dedicated conclusion section with future work is also missing.

  Sources that you should consider adding to your LR:   Mouret JB., Doncieux S. (2008) Incremental Evolution of Animats’ Behaviors as a Multi-objective Optimization. In: Asada M., Hallam J.C.T., Meyer JA., Tani J. (eds) From Animals to Animats 10. SAB 2008.

Lecture Notes in Computer Science, vol 5040. Springer, Berlin, Heidelberg.    "Evolutionary Swarm Robotics: A Theoretical and Methodological Itinerary from Individual Neurocontrollers to Collective Behaviors", Vito Trianni, Elio Tuci, Christos Ampatzis, 2014, The Horizons of Evolutionary Robotics, 153-178.  

Doncieux S., Mouret JB., Bredeche N., Padois V. (2011) Evolutionary Robotics: Exploring New Horizons. In: Doncieux S., Bredèche N., Mouret JB. (eds) New Horizons in Evolutionary Robotics. Studies in Computational Intelligence, vol 341. Springer, Berlin, Heidelberg.    The horizons of evolutionary robotics, 2014, PA Vargas, EA Di Paolo, I Harvey, P Husbands, MIT press.        

Author Response

The text is well organised, contributions are relevant and methods are adequate. However, the proposal was made based on a limited literature review. This must be extended to describe not only the algorithms employed on related work but also the experimental scenarios used and how did they motivate the experiments proposed in this work. Results and analyses from other studies should also be discussed.

We expanded the discussion of the motivations behind the selection of the proposed scenario and the discussion of the related studies. We expanded the list of related paper included in the references. 

I would also suggest a more detailed analysis of the evolution process, including the average fitness over time. Besides, the box-plots used to present the results could be summarised on a single figure, allowing a more straight-forward comparison between the different architectures considered. Some outliers should also be removed for better clarity.  

We included additional figures showing the fitness over time.

Overall, I expect a more in-depth analysis of state-of-the-art, methodology and results for the paper to be accepted as a contribution to this journal. Not only on the motivation for the work carried out including a more comprehensive literature review, but also on the analysis and presentation of results. A dedicated conclusion section with future work is also missing.

We expanded the coverage and the discussion of the related literature. Moreover, we added a concluding section and a discussion of relevant future works.

Round 2

Reviewer 1 Report

The revision of the manuscript has helped improving the overall quality. I understand it would take time to do quantitative comparison to the other approaches and thank you for putting that in future work. The consolidation of result figures into two is somewhat helpful. I do have some major comments:

1. It seems the take-away message of the paper is that MO-ES is better than SSA, MO-SSA, and ES, because it gives statistically significant (p-value<0.05) higher fitness value in two problems? If that is the case, I suggest making the message explicit in Abstract and Conclusion. For example, the paragraph in conclusion:

“The obtained results demonstrated that the method is effective and produce significantly better results than the standard method and a multi-objective method that operates by selecting the individuals located on the first pareto-fronts.”

may be expanded to including the name(s) of the algorithm(s) to be more specific about what you compare.

2. I still struggle understanding the numbers in the text and the numbers in Figures 2 and 3. The two figures seem to be boxplots. If they are indeed boxplots, the use of the word “average” in boxplots needs clarification. Are you referring it to the mean/average of 16 trials for each condition? If it is the mean/average value, then bar charts with error bar are more appropriate, as boxplots show median, percentiles and outliers.

3. The new Figures 2 and 3 show different values from Figures 2-5? Would you be able to share fitness values for all the 16 trials for each condition with supplementary result data?

4. I suggest to take the following paragraph out of the conclusion because it is the given and it is not related to or supported by experiments and results in the paper:

“The method proposed present advantages with respect to alternative methods investigate in the literature. In particular, unlike modular approaches, it does not impose constraint on the architecture of the neural network controller. Moreover, unlike incremental approaches, it does not require the manually shape the course of the evolutionary process.”

Author Response

The revision of the manuscript has helped improving the overall quality. I understand it would take time to do quantitative comparison to the other approaches and thank you for putting that in future work. The consolidation of result figures into two is somewhat helpful. I do have some major comments:

Thank you for your additional constructive comments.

It seems the take-away message of the paper is that MO-ES is better than SSA, MO-SSA, and ES, because it gives statistically significant (p-value<0.05) higher fitness value in two problems? If that is the case, I suggest making the message explicit in Abstract and Conclusion. For example, the paragraph in conclusion:

“The obtained results demonstrated that the method is effective and produce significantly better results than the standard method and a multi-objective method that operates by selecting the individuals located on the first pareto-fronts.”

may be expanded to including the name(s) of the algorithm(s) to be more specific about what you compare.

done

  1. I still struggle understanding the numbers in the text and the numbers in Figures 2 and 3. The two figures seem to be boxplots. If they are indeed boxplots, the use of the word “average” in boxplots needs clarification. Are you referring it to the mean/average of 16 trials for each condition? If it is the mean/average value, then bar charts with error bar are more appropriate, as boxplots show median, percentiles and outliers.

The boxplots show the distribution of performance over the 16 replications of the experiments performed in each condition. Since the robots are evaluated for 4 episodes, 2 on the first behavior 1 and 2 the second behavior, each of the 16 values represent the average of the performance over episodes. We revised the figure caption to make it clearer.

We included also the results for each replications the average and standard deviation in two additional tables (Table 1 and 2) (see the replay to your comment below). 

  1. The new Figures 2 and 3 show different values from Figures 2-5? Would you be able to share fitness values for all the 16 trials for each condition with supplementary result data?

The new Figures 2 and 3 show the same data of the old Figures and additional data. The additional data are the performance obtained on each of the two behaviors.

We included two tables with the performance obtained on the two behaviors on each replications in Table 1 and 2. The tables also include average and standard deviation.

  1. I suggest to take the following paragraph out of the conclusion because it is the given and it is not related to or supported by experiments and results in the paper:

“The method proposed present advantages with respect to alternative methods investigate in the literature. In particular, unlike modular approaches, it does not impose constraint on the architecture of the neural network controller. Moreover, unlike incremental approaches, it does not require the manually shape the course of the evolutionary process.”

Removed as requested.

Reviewer 3 Report

Clearly, there was some improvement following the reviewer's comments, however, the paper still lacks a more in-depth discussion of related work. This must be presented in more detail, not only mentioned or referenced. Simply stating that "several other studies investigate the role of modularity in classification and regression problems and in the context of genetic regulatory networks" might suffice for a conference paper. However, for this journal, the reader would expect a wider contextualization, possibly with the introduction of a new section focused particularly on this aspect. The references must be described and discussed, in order to establish relationships between related work and the developments of your research.

When pointing out the adaptive problems chosen for the research, the paper must provide a comparative discussion on other approaches in the literature that were introduced for this same aim. Which other adaptive problems were introduced as evaluation scenarios for other approaches related to yours?

The methods and results sections were also improved, however, I still miss a more detailed analysis that would justify the publication in this journal. Figures 2, 3, 5, and 6 might be combined as one single figure with four subplots. Figures 4 and 7 also address the same issue and must be combined. Besides, the plots relate to average measurements, hence must be sown together with the standard deviation.

Author Response

Thank you for your comments. We feel that the revised version is significantly improved. 

Clearly, there was some improvement following the reviewer's comments, however, the paper still lacks a more in-depth discussion of related work. This must be presented in more detail, not only mentioned or referenced. Simply stating that "several other studies investigate the role of modularity in classification and regression problems and in the context of genetic regulatory networks" might suffice for a conference paper. However, for this journal, the reader would expect a wider contextualization, possibly with the introduction of a new section focused particularly on this aspect. The references must be described and discussed, in order to establish relationships between related work and the developments of your research.

We added an extended description of the relation of the two algorithms proposed in our manuscript with related works. In particular, we discussed the relation with other evolutionary robotics studies that used multi-objective algorithms and with other works that used evolutionary strategies with multiple gradients.

We did not expand the review of works using modular networks and incremental evolutionary algorithms since they constitute alternative methods to evolve multiple behaviors but are not related with our approach. As we stated in the introduction, the problem of the developing multiple behavior can be approached in three different manners: (1) by using modular neural controllers, (2) by using a form of incremental evolution in which the behaviors are trained one at a time, and (3) by using extended algorithms that force the retention of variations that are adaptive with respect to each behavior (instead than simply the variations that maximize the fitness). In our manuscript we explore the third approach. So, the relation of our work with respect to modular studies [5-14] and incremental studies [15-17] consists in the fact that we use an alternative method.

We could explain the difference between the different modular or incremental studies. For example, we can specify that in [5] and [6] modularity is achieved by using alternative motor neurons and selectors neurons that determine when the first of the second motor neuron is responsible for controlling the actuator. We can explain that [5] use a fixed architecture while [6] use an architecture that vary during the course of the evolutionary process. We can describe the regression problems and the genetic regulatory network problem studied in [10-12]. For example, we can specify that the problem considered in [10] consists in recognizing whether the stimuli presented in the left or right portion of 8-pixel retina are the same or not. However, we think that reviewing this works in more details will not help the reader to better understand the relation with our work. It might rather distract the reader with details that are not relevant with respect to the work presented.

When pointing out the adaptive problems chosen for the research, the paper must provide a comparative discussion on other approaches in the literature that were introduced for this same aim. Which other adaptive problems were introduced as evaluation scenarios for other approaches related to yours?

We included a description of the adaptive problems used in related studies.

The methods and results sections were also improved, however, I still miss a more detailed analysis that would justify the publication in this journal. Figures 2, 3, 5, and 6 might be combined as one single figure with four subplots. Figures 4 and 7 also address the same issue and must be combined. Besides, the plots relate to average measurements, hence must be sown together with the standard deviation.

We combined the figures as requested. In the new figure we show both the average performance on the two behaviors and the performance during the production of each of the two behaviors. We now show the standard deviation in the figures plotting the variation of fitness during evolution by using shadows, as requested.

Round 3

Reviewer 3 Report

The extended background is adequate to contextualize the reader.  It also now states the problem clearly. The new figures in the results section provide a better understanding of the contributions achieved and allow a comparative discussion on the different methods presented. 

Author Response

The extended background is adequate to contextualize the reader.  It also now states the problem clearly. The new figures in the results section provide a better understanding of the contributions achieved and allow a comparative discussion on the different methods presented. 

Thanks you for your feedback. We feel the manuscript is significantly better as the result of the revision.